# Drought Influences on Food Insecurity in Africa: A Systematic Literature Review

**DOI:** 10.3390/ijerph17165897

**Published:** 2020-08-14

**Authors:** Bethuel Sibongiseni Ngcamu, Felix Chari

**Affiliations:** 1Public Management and Leadership Department, Faculty of Humanities, Nelson Mandela University, Port Elizabeth 6031, South Africa; 2Faculty of Business Sciences, Walter Sisulu University, Buffalo City Campus, East London 5247, South Africa; charifelix93@gmail.com

**Keywords:** climate change adaptation, drought, food insecurities, hermeneutic framework, resilience, traditional leadership

## Abstract

African countries continue to be prone to drought, caused mainly by unfavorable weather patterns and climatic variations which have an adverse impact on rural households and agricultural production. This literature review article accounted for the aforesaid drawbacks and attempted to assess the effect of drought on food insecurity in African countries. This article further sought to dissect the resilience and climate change adaptation strategies applied by African countries to mitigate the adverse effects of drought on food insecurity in rural livelihoods. The hermeneutic framework was adopted in this study, where the secondary data sources were searched from credible bibliographic and multidisciplinary databases and organizational websites. Thereafter, it was classified, mapped, and critically assessed using the qualitative data analysis software NVivo to generate patterns and themes. The NVivo program is a qualitative data analysis software package produced by QSR International and which helps qualitative researchers to organize, analyze, and find insights in qualitative data; for example, in journal articles where multilayered analysis on small or large volumes of data are required. This article has the potential to contribute in theory, concept, policy, and practice regarding best practices, resilience, and climate change adaptation strategies that can be harnessed by rural people. Furthermore, this article has the potential to shed light on the role played by traditional leadership and policy improvements in ensuring there is sufficient food during periods of drought.

## 1. Introduction

African populations that mostly live below the international poverty line have been affected by growing undernourishment, famine, malnutrition, hunger, and the extreme manifestation of acute food insecurity at an unprecedented rate. The extreme food shortage has been attributed to a combination of factors including prolonged droughts, adverse weather patterns, civil wars, political-economic challenges, and diseases (such HIV/AIDS and COVID-19) as well as poor governance and government policies. The frequency, duration, and intensity of droughts have generally increased worldwide, posing a constant threat to world food security. More than 11 million people have died and more than 2 billion have been affected worldwide as a result of droughts since the turn of the century—more than any other physical hazard (World Meteorological Organization). [1] posit that agricultural countries, such as Niger, employ over 80% of the country’s workforce, with an increasing number of the poorest of the poor in rural areas depending on unreliable and erratic rainfall patterns and recurrent droughts. This results in recurring deficits of production which generate food insecurity and exacerbate poverty. [2] further argues that drought diminishes agricultural outputs, hampering economic growth, employment levels, food security, and poverty reduction [3]. In addition, [4] also observes that erratic rainfall has diverse effects on food security in rural communities that rely on subsistence economies. Given these statistics and published data that have been attributed to drought, this study set out to provide an in-depth synthesis of published literature. In addition, it aimed to establish dominant themes on the effects of drought on food security, adaptation, resilience strategies, and vulnerabilities in rural areas in Africa to inform national and international policies. This was triggered by a host of empirical studies that noted a paucity of research on the relationship between drought and households’ food insecurity, and the measurement of households and food insecurity during and in the aftermath of drought [5]. There is also a shortage of data in sub-Saharan Africa on the association between drought and violence against women [6] and between food insecurity and racial and gender conflicts [7].

It is also notable that droughts are common in the leading economies in Africa, including the northern parts of Nigeria, which is considered a food-deficit country [8]. In South Africa, another country with a leading economy in Africa, drought has had a devastating effect on agricultural production, costing farmers millions of rands [9]. As a result, there have been protests against food insecurity by civil society groups fighting for food justice and the right to food [10].

Other researchers [8], meanwhile, believe that the prevalence of malnutrition and people who experience food insecurity largely occurs in rural areas, exacerbated by inadequate infrastructure that influences poverty [8]. Philipose [10] suggests that the consequences of drought mostly in rural communities are due to donor tensions and government corruption, resulting in people’s livelihoods being negatively affected, reduced incomes, food insecurity, and political problems. The author further opines that the HIV/AIDS epidemic also leads to high poverty levels, unstable and higher food prices, reduced incomes, and significant food insecurities. Job security for unemployed rural dwellers is dwindling as a result of the scarcity of job opportunities [11]. Furthermore, poor infrastructure (mostly roads) and technology [12] has led to the transport of goods in or outside of rural areas becoming more expensive [12]. Consequently, in the past two decades, a host of recent studies have been conducted on the detrimental effects of drought on different countries’ economies and on commercial agricultural production. However, there is minimal empirical evidence on the effects of drought on subsistence farmers and rural households, which are mostly dependent on the land for their daily survival. Furthermore, as shown above, there is an existing gap in knowledge on rural householders’ adaptation and resilience strategies in response to the effects of drought and ensuring there is sufficient, balanced, and nutritious food for rural inhabitants during drought periods. Stemming from the aforementioned synthesized literature, there is an existing void due to the paucity of empirical and scholarly studies on governments’ contingency plans in preparing rural communities to be food secure in times of drought. In addition, there is limited literature on the role of public–private partnerships and the blending of indigenous and scientific knowledge in curbing food insecurities during such devastating periods. An attempt was made in this study to demonstrate the gap in the existing strategies on climate change adaptation, disaster risk reduction, and resilience to drought in terms of food security in rural households. The void identified has been worsened by conflicting definitions of the various concepts that influence food security.

There is a myriad of variables that are central to this study, which include drought, food security, adaptation, disaster, resilience, and climate change, with unique definitions and interpretations. For instance, drought is regarded as a prolonged period where the normal rainfall drops below the average and which results in a decline of water in reservoirs, a decrease in stream flow, and damage in crop plantations. Drought is broadly defined as “a deficiency of precipitation over an extended period, usually a season or more, which results in a water shortage for some activity, group, or environmental sectors” [13]. The Rome Declaration of World Food Security (1996) terms food security as constant and sufficient access to nutritional, safe, and quality food to meet the needs of the members of the family [14,15]. According to World Food Summit of 1996, “food security exists when all people, at all times, have physical, social and economic access to sufficient, safe and nutritious food that meets their dietary needs and food preferences for an active and healthy life”. Marshall [16] defines food “insecurity as the inability of livelihoods to guarantee access to sufficient food at the household level”. Meanwhile, a host of researchers define climate change adaptation differently [17]: As an adjustment in response to natural and human systems as a result of climate change, as a significant supplement to mitigation [18], and as a process that necessitates farmers to identify and implement adaptation strategies due to climate variations [19]. Resilience as another pertinent term in this study is understood [20,21] to mean to adjust, recover from, and overcome the impact of disasters. The existing and available literature on the definitions of the variables of this study suggests that there are no universal definitions. This is demonstrated by the declarations of the United Nations (UN) and a host of other authors having multiple perspectives concerning these concepts. This suggests that these concepts can be better understood and defined in a particular environment and situation, which invites various authors to explore further concepts to suit their respective geographies.

Against this backdrop, this literature review article sought to close the gap in the body of knowledge by analyzing previously published knowledge on the influence of drought on food insecurity, vulnerability, and coping strategies in rural African communities. This article has the potential to contribute to this sparsely researched subject (drought and food insecurity) in theory, concept, policy, and practice regarding best practices, resilience, and strategies in order to reduce the vulnerability of agricultural systems to climate change, thus alleviating food insecurity in Africa. This literature review study intended to identify gaps in the existing data and identify priorities for future research. 

## 2. Materials and Methods

This synthesized review of the literature was planned, implemented, and systematically recorded according to the chosen framework. A hermeneutic framework was adopted in this study, integrating the analysis and interpretation of, and the search for, literature [22]. Secondary data sources were searched, classified, mapped, and critically assessed to generate themes and arguments. This literature review article analyzed previously published knowledge on the impacts of drought on food security in rural communities in African countries. The review study considered articles covering Africa that were published in peer-reviewed journals and that measured the effects of drought, vulnerability to food insecurity, and adaptation and resilience strategies. 

## 3. Search Strategy

Both researchers were assisted by a librarian in the search strategy development. A systematic search was conducted between 6 January 2020 and 31 March 2020. Electronic bibliographic databases and databases from different disciplines and websites—such as Africa Journal Online, Applied Social Online Index and Abstracts, Food and Agriculture Organization (FAO), Google Scholar, World Bank, Library, Organisation for Economic Co-operation and Development (OECD), United States Agency for International Development (USAID) and World Vision—were used. The study used the following keywords: Drought in Africa, food security in Africa, effects of drought in Africa, vulnerability to food insecurity in Africa, climate change adaptation, and resilience strategies. Common themes were identified and extracted to generate insight into food insecurity in drought situations.

A Microsoft Excel spreadsheet was developed where all the retrieved materials were recorded for further screening, analysis, and screening assessment. Research studies in Africa were selected, where the majority of cases were rural and prone to drought and food insecurity. Only those peer-reviewed articles between the years 2000 and 2020, technical reports, conference papers, declarations, and manuscripts that were authored in English and were accessible (without any financial costs) and with the main themes of drought and food security were included. An Excel spreadsheet was perused to identify relevant studies that meet the current literature review objectives. Some studies were excluded, such as those published before the year 2000, as well as letters to the editor and articles that were not authored in English and had no connections to drought or food insecurity in Africa. The selected articles were transferred to NVivo, a qualitative analytical tool, for further analyses, coding, categorization, and development of themes. This took the form of nodes, which were used to generate relevant patterns and theme.

## 4. Results

This section critically synthesizes the literature review on four themes which are central to this study. These themes include the adaptation strategies used by local farmers in Africa, drought influences on food insecurity, the effect of drought on vulnerable groups, and resilience strategies.

### 4.1. Study Characteristics

This literature review found a total of 26 sampled relevant empirical studies out of a total of 1346 studies that met the objectives of this study on drought influences on food insecurity in Africa. A total of seven studies used the qualitative research approach and method, 10 used the quantitative method, seven used a mixed methods approach and only two used a literature review study. The study sample was comprised of countries in sub-Saharan Africa (including Malawi, South Africa, and Zimbabwe) with some in the western region (including Benin, Ghana, Niger, and Nigeria). The majority of the sampled studies focused on small-scale farmers in rural areas with the remaining studies focusing on the country as a whole. The majority of the studies’ objectives focused on the impact of drought on food insecurity on local rural communities and farmers, as well as adaptation strategies in agricultural production [8,11]. Very few studies focused on the utilization of technology associated with food security and growth output in the agricultural sector [8]. Nonetheless, all studies were aimed at developing a relationship between drought and the effects of food insecurity in rural households. Some of the studies cited in this article are relevant to this study as they support sampled empirical studies that are listed in Table 1. Some of the supporting articles not featured in the table did not fully meet the initial selection criteria of research focusing on drought influences on food insecurity in local communities in Africa. 

### 4.2. Effects of Drought and Vulnerability on Food Insecurity: Vulnerable Groups

Philipose [10] cites the indirect impacts of drought on the most vulnerable groups, including female-headed households, whose income is derived mostly from agriculture. This has been compounded by low education and literacy and poor health among women, exacerbated by HIV/AIDS [38]. Philipose has also stated that in the 19th century, environmental changes in Africa have directly affected vulnerable groups, including children, people living with disabilities, ethnic minorities, and the aged, all of whom depend solely on nature for their daily survival [39].

Numerous authors [1,23,29,40,41] reveal the impact of drought on rural households and livelihoods. These authors attribute such challenges to erratic climatic variations, rising temperatures, and floods. The rural poor in Africa are considered to be dependent on semi-subsistence agriculture for their survival, which is sensitive to changes in weather patterns. According to [1] the agricultural sector is mostly affected by the adverse effects of climate change, with the highest number of rural inhabitants prone to chronic hunger and malnutrition. The magnitude of poverty limits the ability of those affected to adapt to climate variability and natural disasters [42]. Consequently, a number of authors [23,43] cite rural households and communities in sub-Saharan Africa as having battled with socio-economic challenges due to climate change and variability.

The adverse impacts evidenced in this article have been experienced in the first decade of the 20th century. Clover [15] cites the challenges of food insecurity, which are created politically and exacerbated by faulty analysis and actions, a failure to understand the interventions of the World Food Summit News of 2002, political interference, victimization, discrimination by government and a lack of political power and will. This has been prevalent in some African countries, which include Eritrea, Ethiopia, Angola, Sudan, and sub-Saharan countries, where the highest prevalence of undernourishment has been experienced. Moreover, the previous authors contend that countries such as Malawi and Zambia have experienced a decline in food availability during years of drought. This was mostly attributed to a lack of transparency concerning government policy on trading. [44] mention a host of direct impacts of drought on farmers due to a loss of crop production and maize.

Sutcliffe, Dougill, and Quinn [23] examined how government policy on climate adaptation affected farmers, concluding that insufficient government support constrained farmers’ initiatives. [36] blamed food shortages in rural areas of Zimbabwe on poor economic policies and the violent land seizure in 2000. This author further claims that during the drought, politically connected people would buy grain from the Grain Marketing Board and then resell it on the market at higher prices, thereby gaining financially from food insecurity in the country. [30], meanwhile, identifies the souring of donor–government relations as the cause of donors’ slow response to the 2001/2002 drought in Malawi, with food aid arriving late and which led to severe malnutrition and a high mortality rate. According to other researchers [31], poverty results in the unsustainable use of natural resources and the overall deterioration of the environment. These authors noted that during a severe drought in Ekiti State in Nigeria, many of the poor people were forced to leave pastoralism and migrate to towns in search of paid jobs. An increase in farming and non-farming income improved the use of water and soil conservation measures and the likelihood of mixed farming.

According to [33], the main problem encountered by farmers and which leaves them extremely vulnerable and unable to plan is the inadequate weather. [34] made similar observations in Zimbabwe’s Seke and Murewa Districts where a significant proportion of farmers had no access to early weather forecasting information. Six studies [9,10,23,26,28] identified drought as adversely affecting small-scale farmers and government response strategies (see Table 1). Philipose [10] examined how food security had been influenced by drought in selected countries with similar characteristics (Malawi and Zambia) that make them vulnerable to food insecurity, with the designated groups mostly affected by HIV/AIDS. This researcher suggested a solution that included formal and informal trading and innovative programs. Meanwhile, unscientific adaptation strategies observed by smallholder farmers in Malawi on changes in rainfall, including seed network and maize producers [23], had been disputed by accurate meteorological data. Open communication channels and dialogue among stakeholders (including scientists) was recommended in providing accurate data to local farmers. Local farmers in South Africa were found not to have received early warning information on a drought in a study by [9]. Recommendations were made that farmers needed to be trained on drought mitigation strategies, alongside the establishment of warning systems. Strengthening community capacity in managing the effects of drought through preparedness and adaptation strategies would help to reduce the possible threat of food insecurity [45], as well as an investment in Drought Early Warning Systems (DEWS). In addition, [45] argued that DEWS add value to households’ food security: The available and reliable information assisted them in timely planting, the diversification of crops, drought management, purchasing the right farm equipment, and planting crops that are drought-tolerant [45].

The impact of drought in Africa is severe due to the backlog in infrastructure development [46]. In a study in Malawi, [30] emphasized the vulnerability to drought due to the declining access to inputs and infrastructure as a result of population growth and density. This resulted in the reduced cultivation of high-yielding varieties of maize and low yields of staple crops. Numerous articles relevant to drought risks emphasize the importance of infrastructure development in reducing vulnerability to drought [47]. The UN′s Office for Disaster Risk Reduction [48] proposes an investment in agricultural infrastructure and technologies such as the construction of dams, communications infrastructure, and irrigation infrastructure. Irrigation strategy was seen as a viable method that could help to improve crop production in places prone to drought. [23] posit that farm yields in poor climatic conditions can be supplemented by the availability of dams and wells, which can contribute significantly to reducing hunger and assisting in vegetable gardening.

The adverse effects of climatic change variations in African countries are exacerbated by political interference and will and less transparent and responsive government policies. Such geographies that are prone to drought directly and negatively affected vulnerable and indigent groups in rural areas—mostly illiterate, women-headed households that are affected by slow-onset disasters such as HIV/AIDS. The combination of the negative effects of climate change and governments’ incapability has reduced agricultural output and crop production, which has exacerbated chronic hunger and malnutrition, food insecurity, and prevalent undernourishment among poor rural communities. However, there is also limited and compelling empirical evidence to ascertain the impacts of adverse climate change, reactionary strategies by government, and political interference as well as a lack of political will in indigent and vulnerable rural communities in drought and poverty-stricken rural communities. In a similar vein, there are gaps in the previously published data on the extent to which droughts have affected the food security of women, the elderly, and children in disaster-prone areas.

### 4.3. Resilience Strategies

A host of strategies (see Table 1) have been considered as being effective against drought, which include gender, age, adaptive farming, race, community involvement, governance, traditional knowledge blended with scientific knowledge, understanding [48] and learning from risk [27], gap identification [49], resilience standards, and evaluation [21]. Some of these authors further suggest that responsive interventions should take place, with plans developed, implemented, and measured with a clear view to strengthen resilience capabilities in rural communities. Accordingly, the Southern African bloc has been seen as cyclical regarding drought response and contrary to various frameworks that were aimed at strengthening resilience [48,50]. [51] recommends formal or informal institutional arrangements, models, and platforms—such as farming communities and civic groups—that could encourage participation and enhance drought resilience [52].

A plethora of challenges have been highlighted in the Southern African Development Community (SADC) countries, which diminishes opportunities for the implementation of resilience interventions and make it difficult for humanitarian responses to focus on resilience. These mostly include political interference, environmental degradation, institutional incapacity, poverty, and a lack of poverty alleviation policies [21] with humanitarian responses to drought focusing on emergency needs, which are critical in rural communities [53,54]. While drought resilience strategies have been sparsely observed in countries such as Lesotho, Namibia, and South Africa [55,56], negative repercussions brought about by droughts show that there are inadequate resilience measures to inform policies and interventions [21]. However, the previous mishaps can be rectified by building resilient capabilities among farmers and enacting agricultural policies such as funding: Providing farmers with agricultural equipment and educating them on the effective and efficient use of such equipment, thereby assisting them in matters where they may be vulnerable. A number of these innovative policies have failed due to inefficient management [57] with less influence of tribal leaders being factored into such resilience strategies [58]. Despite being considered to be corrupt, rural communities still trust and value these leaders more than elected officials. [21] argue that this situation is worsened by the lack of a contextually and culturally appropriate resilience framework. With community resilience remaining poorly understood, there is little success in attempts to develop local resilience frameworks. This is perpetuated in countries such as Lesotho and Swaziland where the structures of resilience are poorly understood, despite such countries being prone to the recurrence of drought, which adversely affects rural subsistence communities [21,59]. In Lesotho, the resilience framework is non-operational and remains in a draft form. Unlike the research on the impact of droughts on agricultural production and the commercialization of farms, little attention has been paid to measurement, standards, and strategies to counteract the negative impacts of food insecurity on rural communities and subsistence farmers in drought-prone areas. In addition, the integration of traditional and scientific knowledge and its effectiveness and impact have not been tested empirically in areas affected by drought and food insecurity in all regions in Africa. Furthermore, there is a research gap in the published data on capacity levels of governments’ workforces in dealing with the implementation of resilience strategies in rural communities.

### 4.4. The Effects of Adaptation Strategies on Local Farmers in Africa

A total of 19 studies out of 26 [1,4,19,24,25,32,33,37] assessed local farmers’ and communities’ adaptation strategies in the event of adverse climatic changes (see Table 1).

The word count (Figure 1) generated from the NVivo program clearly depicts an association between climate change and its effects on food security in Africa. Figure 1 further reveals that farmers and communities are adversely affected by drought with the adaptation and resilient strategies lacking to avert such crisis. For instance, local farmers were assessed in two Niger districts (Diffa and Aguie) [1] on their knowledge of climate variables as compared to meteorological events and adaptations. The researchers found discrepancies on the meteorological observations from the farmers and suggested that they needed to be equipped with agricultural technological skills and reliable information and have access to crops that are resistant to high temperatures. Selected papers analyzed and evaluated the role of drought-resistant crops and breeds in reducing the effects of drought in rural areas. Crop variety diversification and drought-resistant crops or breeds are highlighted as coping strategies. For example, in a study in Malawi [33], researchers found that Malawians increasingly prefer short-season maize as a strategy for adapting to drought. They, therefore, suggested the adoption of maize cultivars that are tolerant to climatic extremes and variability as one possible solution. A similar perception study was conducted by [24]. They targeted subsistence farmers regarding climatic variations and the socio-economic changes that influenced their agricultural livelihoods. They needed adaptation strategies to overcome extreme and unpredictable temperatures, rainy seasons, unfavorable transportation networks, diseases, and the lack of irrigation infrastructure. These authors concluded that the subsistence farmers had no adaptation strategies in their farming system although there were hidden resilience strategies that could be harnessed. A synthesized literature review on farmers’ perception on climate change [24] and their adaptation strategies found fascinating and unique factors that influence farmers to adapt to climate change in the sub-Saharan region. These include gender, age of the head of the household, experience in farming, household size, education level, and access to credit facilities. This interesting study had implications regarding the public–private partnership policy on farmers, and recommended climate change adaptation policies be included in governments’ development programs, investments in resilience programs, and monitoring and reporting systems. In addition, farmers with a higher level of education and years of formal education have been found by a plethora of researchers [31,33,35] to be able to adapt better to climate change, to be able to diversify to non-farming activities, to adjust planting periods, and to easily utilize agricultural technologies as compared to their counterparts who are less educated.

Consequently, in countries such as Namibia, a study by [2] examined subsistence farmers’ adaptation abilities to climate risks threatening crop production. The community members were found to be dependent on their indigenous knowledge of rainfall prediction. Meanwhile, the same author in 2016 examined the effects of the scarcity of rainfall on the subsistence crop production and the significance of religious rituals as adaptation measures where cultural knowledge was used by women as an adaptation strategy. As a drought mitigation strategy, farmers have been recommended to rely on indigenous knowledge. Indigenous knowledge systems in countries such as Zimbabwe are used to predict weather [36] and in farming practices to reduce disaster risks [60]. In the Niger Republic, [1] observed that crop diversification strategies have mostly been used traditionally to reduce risks and adapt to climate change as a result of the prolonged rainfall variability. In addition, another researcher [61] found that northeastern Ghana farmers planted multiple indigenous drought-resilient crop varieties to cope with drought and, subsequently, food insecurity. The reviewed literature above shows a significant gap in adaptation strategies to climate change because its effects have not been felt by local farmers. Researchers have not yet been able to espouse the strengths of indigenous strategies as they have not been harnessed.

### 4.5. Drought Influences on Food Insecurity in Africa

The impacts of drought go beyond agriculture into related sectors, affecting major food supply chains and resulting in episodes of price spikes. This is because of the reliance of these sectors on agriculture for raw material so as to increase food production, which is needed to provide for the ever-increasing population [62]. According to Kebede, Atlin, and Melchinger [63], it was drought that led to the shortage of supplies, which dramatically increased food prices and substantial reliance on imports. This had a direct and severe effect on the more vulnerable and poor communities.

The combination of the socio-economic and environmental issues has resulted in food insecurity that negatively impacted the well-being of the people, the economy, and the environment [64]. For instance, in Southern Africa, dam levels have reduced severely, leading to a reduced water supply and poor quality of water. This, in turn, led to crop failure, a reduction in production and diminished power generation [65,66]. Hailea et al. [65] found that in East Africa and in the Great Lakes Region, the availability and quality of water suffered as a result of drought, affecting socio-economic activities such as agriculture. In a study in Ghana, researchers [23] established that most farmers experienced the adverse impact of drought, which included a shortage in the water supply for both humans and livestock. Various researchers have shown the adverse consequences of drought; for instance, Sutcliffe, Dougill, and Quinn [23] highlighted the fact that drought has a negative effect on unemployment as farmers lose their alternative source of income. In addition, there is a reduction in growing seasons and in agricultural yields due to the reduction in the area suitable for agriculture [67]. Meanwhile, Kebede, Atlin, and Melchinger [63] noticed people’s inability to grow food and rear livestock. Numerous authors have recommended strategies to reverse the above, such as farmers diversifying in the dry season, for example, beekeeping, rearing livestock, weaving, and dry season gardening [33]. Instead of the temporal rural–urban migration in search for non-existent jobs, the sale of fuel, wood, and charcoal could be considered to be diversification adaptation strategies for their livelihoods, as suggested by Kumasi, Antwi-Agyei, and Obiri-Danso [32]. The reviewed literature above on the effects of drought in Africa depicts anecdotal information that is not supported or validated by empirical evidence. The majority of the studies are qualitative and focus on the country in general rather than the most affected areas. These tend to be rural communities and, therefore, the well-being of the poor is most affected, increasing food insecurity. This suggests a serious gap in the literature, which requires further exploration by researchers.

## 5. Conclusions

This article considered the literature of the past two decades on the influence of drought on food insecurity in rural areas in Africa and which espoused four fascinating themes: (1) Climate change adaptation strategies, (2) the effects of drought on food insecurity in Africa, (3) the vulnerability to food insecurity due to drought, and (4) local resilience strategies. The literature reviewed in this article highlighted a number of gaps in the existing data published between the years 2000 and 2020. However, significant gaps have also been observed in this review. These include the fact that rural households and subsistence farmers’ indigenous knowledge as adaptation and resilience strategies has been overlooked as a solution to persistent and recurring droughts and the effects on food insecurity, as well as the potential benefits of blending traditional and scientific knowledge. In addition, anecdotal evidence has been observed on the effect of drought on the designated groups in rural communities, with minimal empirical studies conducted to ascertain and determine the local resilience strategies used by them. Lastly, interventions with multiple stakeholders, including governments, have not been evaluated by researchers, which is a significant void in the literature.

Consequently, mitigation against the effects of drought requires a multi-strategy and multi-stakeholder framework in order to ensure resilience in food security. The strong cooperation among key stakeholders—including indigenous leaders, communities, civil society groups, and government—is a valuable asset in monitoring and evaluating drought. Based on this literature review, a conceptual framework is suggested that integrates the collaboration of stakeholders and all the different strategies in order to mitigate the effects of drought on food security. In addition, this synthesized literature review highlights the effect of drought on the rural poorest of the poor. It poses a serious threat, adversely and consistently affecting people’s nutritious food security, a situation which is prevalent in sub-Saharan Africa. Moreover, rural communities and farmers’ survival strategies and their effectiveness on food security during and in the aftermath of drought periods, which are informed by communities’ cultural and indigenous knowledge, have not been empirically tested and assessed by scholars, which suggests future exploration. There is also a significant gap in knowledge on the effects of drought on individual households and designated groups’ adaptation and resilience strategies on food security, which focuses mostly on small- and large-scale farmers. Nonetheless, researchers and scholars have paid limited attention to blending traditional knowledge, local resilience strategies, and scientific knowledge. Most of the rural farmers believed in the traditional structures and systems as adaptation strategies, which necessitates future researchers to explore these untapped strengths and weaknesses. The measurement and evaluation of the impacts of government interventions provides a significant gap for future researchers to explore. This includes the communication with, and training of, small-scale farmers and rural householders on drought adaptation and resilience strategies in order to ensure abundant food security.

## Figures and Tables

**Figure 1 ijerph-17-05897-f001:**
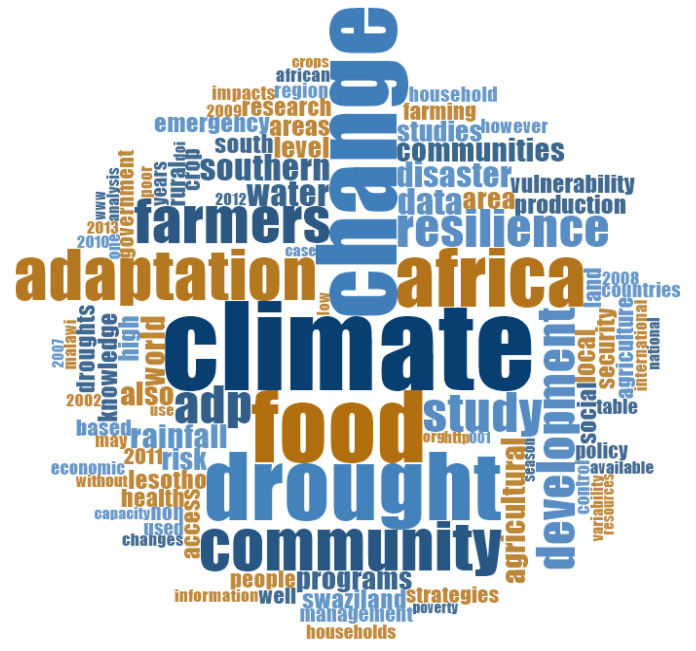
Word cloud. Source: Authors’ own creation.

**Table 1 ijerph-17-05897-t001:** Characteristics of the study.

Article Authors	Research Design and Methodology	Sampling and Setting of the Study	Research Objectives	Themes	Conclusions/Impacts of a Specific Drought and Vulnerability to Climate Change
[10]	A qualitative study analysing policies on governments’ response to drought crisis	Two similar countries in characteristics (Malawi and Zambia) in the SADC region	Examine the impacts of drought on food security and response strategies in the Southern African crisis in 2001 and 2002	Both countries have similar features which render them vulnerable to food insecurity, are dependent on maize, have similar patterns of weather and have poor health standards. The prevalence of HIV/AIDS adversely affects the populations including people without land, households headed by women, and orphans	Policy recommendations include increasing food reserves, encouraging formal and informal trade, establishing social protection and innovating programmes, and establishing a global contingency fund. The drought effects on increasing food insecurity have necessitated a myriad of organisations such as the World Bank (WB), World Food Programme (WFP) to provide a humanitarian assistance, the United Kingdom’s Department for International Development (DFID) and Oxfam. These organisations offer support and create safety net programmes
[11]	A qualitative literature study	n/a Generic challenges faced by those living in rural areas in Malawi	Examine the challenges faced by the rural poor of Malawi; address important issues and suggest recommendations on the development of national and rural policies	In Malawi, the liberalisation of policies and market interventions are failing; the market has constrained the existing intensive and productive technologies that suit local agroecology	Effective rural development strategies and policies are important for Malawi to develop economically. Also important are understanding the relevance of the study on the process of developing rural areas in Africa; learning from other countries in local and national institutions regarding agricultural technologies and opportunities, communications and infrastructure; and non-farming opportunities. Intensive maize technologies are suggested. The food crisis in the aftermath of drought in Malawi has had an impact, drawing stark attention to the failure of development policies which have resulted in a low return to farmers and service providers’ investments
[1]	The qualitative and quantitative research approaches used in this study were gleaned from the daily meteorological records of temperature and rainfall in two selected areas; simple random and multi-stage procedures were performed	*n* = 284 (Diffa district) *n* = 98 (Aguie district) farmers Farmers in Niger	The meteorological events, as well as adaptations and access to agricultural extension services, were tested on local farmers	Both areas have climatic challenges; crop diversification was adopted as an adaptation strategy	Training and development programmes for farmers equipped them to adapt to extreme temperatures. A traditional risk reduction and adaptation strategy to address the long-standing inter-annual and intra-annual rainfall variability in the area—instead of the climate change adaptation strategy—was considered to be unresponsive to the perceived weather patterns
[23]	Both qualitative and quantitative meteorological records were analysed	Semi-structured interviews and questionnaires Ngabu and Kasungu-Malawi	An understanding of the maize producers (smallholders) was assessed in Malawi. An assessment was also extended to seed network stakeholders at a national level. The main objective was to determine their understanding of changes in rainfall patterns and decision-making regarding choices of maize cultivar	The farmers believe that climate change has led to the seasons growing shorter, motivating them to prefer short-season maize, which is considered to be a good adaptation strategy to drought	The disagreements on the findings recommend a dialogue amongst stakeholders, including farmers, the government and meteorologists. Communication and understanding should be enhanced to address the partial knowledge within seed systems which might assist in achieving agricultural adaptions to climate change
[24]	Qualitative research approach	*n* = 59 households (semi-structured) *n* = 60 focus group Ghana Selected areas of interest were Amponsakrom No. 2, Mantukwa and Meta	A perception study was conducted with smallholders in three selected areas concerning their agricultural livelihoods and adaptation strategies to climatic and socio-economic changes	Extreme temperatures and delayed and unpredictable rainy seasons have directly influenced farming systems in these zones. The socio-economic activities that have affected communities include transportation networks (e.g., roads), farming inputs along with exorbitant prices, diseases and crop pests, and a lack of irrigation infrastructure	There were hidden resilience strategies which were not harnessed by farmers; these had the potential to strengthen their adaptation strategies. The smallholder farmers had no strategies in place to help them adapt their systems of farming
[25]	Qualitative empirical literature review	n/a sub-Saharan Africa	A literature review was conducted in sub-Saharan Africa on the views of farmers on climate variations, its effects on their livelihood and adaptation strategies	The decision on climate change adaptation was influenced by the age of the household, gender, farming experience, size of the household, education level, access to credit facilities, distance from the market, extension of services access and off-farm generating activities	Different countries are prone to a myriad of climate change risks, resulting in a comparative synthesis on different studies on the impacts, perceptions and adaptations of countries in sub-Saharan Africa. The findings have implications in different farming sectors on public and private policies. Adaptation policies should be incorporated by governments in their development agenda. Investment in the resilience strategies is recommended, such as the construction of water infrastructure and monitoring and reporting stations, which would improve the existing knowledge and improve the response to climate change
[26]	Quantitative A secondary epidemiologic analysis was performed	Eastern and southern Africa	In the eastern and southern parts of Africa, malnutrition trends of children were determined in the short-term and long-term, as a result of drought and HIV	National average level of nutrition in children improved	When drought is absent, the percentage of malnourished children improved
[27]	Quantitative	Southern Africa A myriad of data collections tools were used, including interviews with stakeholders, questionnaires, focus groups and documentary historical and archival assessment Key informants included agriculture officials from the Southern African Development Community (SADC) region as well as Natural Resources officials and the SADC Secretariat	Factors adding to drought vulnerability in the SADC region are associated with the inability to understand, integrate and learn from the previous interventions and reactive governments’ disaster risk reduction, informed by a “bail-out” mentality	This study concluded that learning from past and present drought events is a catalyst for adaptation practice	
[28]	Qualitative participatory methodological tools were used	*n* = 40 (Shorobe) *n* = 63 (Tubu) *n* = 28 (Xobe) 18 focus group Botswana	Residents from the Okavango wetlands were targeted, where local knowledge was ascertained	Local institutions of governance were embedded with community adaptation strategies in the aftermath of the human and animal diseases outbreak as well as the recurring drought which adversely affected the livelihoods and well-being of communities	Historical knowledge possessed by local communities on environmental change and adaptation should be incorporated during policy or programme formulation processes. People of different ages are important in crafting comprehensive adaptation strategies that incorporate historical memory
[2]	Qualitative	*n* = 100 North Central Namibia	An ability for communities to adapt to climate change was conducted in North Central Namibia	There are limitations to crop production’s impacts by climate hazards	
[8]	Quantitative descriptive, empirical and econometric techniques were used, and a multi-econometric method	In Nigeria, an analysis was performed on the econometric technique based on the Autoregressive Distribution Lag framework	The extent to which agriculture and the level of technology improve technical knowledge in Nigeria was examined, with the aim of achieving food security	To solve food insecurity challenges, the availability of arable land was a solution; collaborations and interactions between farmers and government—relating to planning issues on food production—can yield positive results	A myriad of factors has been mentioned to improve food security in Nigeria. This includes the establishment and application of an institutional framework, technology, the availability of arable land, interactions between government and farmers, reducing corruption, and improving the electricity supply
[29]	Qualitative report	Lesotho and Swaziland	A resilience study was conducted to demonstrate how best to leverage development and humanitarian interventions, based on a resilience framework that promotes partner communities’ absorptive, adaptive and transformative capacities	This report challenges and shifts beliefs that are widely held on the perception that associates resilience to a particular sector or a stand-alone intervention. A multisectoral approach is required to contribute to community resilience in order to collectively and mutually reinforce positive outcomes	
[30]	Qualitative analysis	Malawi	The influence of safety nets on both protection and promotion effects	Livelihood promotion and -protection	Every income transfer was invested in activities that are income-generated
[31]	Quantitative	Questionnaires	To establish farmers’ adaptation strategies to ensure food security and poverty alleviation	Poverty reduction and food security	The results reveal factors that affect farmers’ choice of adaptability strategies
[9]	Quantitative	*n* = 85 beneficiaries semi-structured questionnaire South Africa North West	To assess the state of drought preparedness by farmers who were the recipients of a cattle developmental project	Farmers’ drought preparedness was hindered by inadequate early warning information and a shortage of funds	A recommendation was made on the plausible contacts to the agricultural extensionists; farmers trained on effective drought mitigation strategies and the dissemination of early warning information is suggested
[19]	Quantitative Zou Department Benin	*n* = 120	To understand the determinants of farmers’ adaptation strategies	Perception of climate change by farmers have been developed; strategies in response to climate change have been adopted by farmers	Solutions and opportunities need to be provided to farmers to allows them to adapt to climate variations
[21]	Qualitative	*n* = 176 (17 focus groups) Lesotho and Swaziland	To enhance resilience in two countries, a research study was conducted examining the influence of drought on local communities’ resilience strategies	Changes in behaviour and knowledge in adapting and applying appropriate actions influenced resilience activities; the effectiveness of institutional support is linked to harnessing a community’s knowledge, their interactions and their involvement in decision-making	It is recommended that resilience within rural communities be enhanced. Large-scale interventions are needed to maintain cohesiveness, in addition to policies that integrate resilience and urgent national development planning
[32]	Both qualitative and quantitative	*n* = 57 and 143 *n* = 400 questionnaires focus group Two districts in Ghana were sampled	Small farmers’ adaptation strategies were identified	Farmers adapted to climate variation and variability	Female farmers have more challenges than their male counterparts
[33]	Both qualitative and quantitative	*n* = 330 small-scale farmers *n* = 150 focus group Ghana	The study assesses adaptation strategies and constraints hindering farmers from being able to adapt	The majority of the countries became vulnerable to climate change variations due to poor agricultural support from government, extreme weather and poverty	Key aspects were access to weather information and capacity development of farmers to climate change adaptation
[34]	Quantitative	*n* = 300 communal farmers Zimbabwe (Seke and Murewa Districts)	Establish the effects of weather forecasting and early warning information systems	Vulnerability to droughts	There was no access to timely and reliable information on droughts
[35]	Quantitative	Ethiopia	Studying the rationale for repeated ploughing by farmers	Repeated tilling and productivity	Disturbance occurred on unploughed strips of land which were left between adjacent furrows, caused by repeated ploughing
[36]	Both qualitative and quantitative	Semi-structured questionnaires, interviews in Bohera and Chikomba Districts of Zimbabwe	An assessment of respondents’ degrees of vulnerability to the effects of drought	Vulnerability to disaster	Weather predictions were conducted by communities through their indigenous knowledge systems. Community members employed their indigenous knowledge to predict weather patterns
[8]	Quantitative	Nigeria	Influence of technology on food security	Food security	The use of technology enhances productivity
[2]	Qualitative	*n* = 50 focused group on women	Research was conducted with women on the effects of a shortage of rainfall on crop production; religious rituals were assessed as an adaptation measure to unpredictable rainfall	Women draw cultural knowledge as subsistence crop producers; various adaptation strategies such as seed-dressing, rainmaking and crop maintenance rituals, which women depended on, ensured food self-sufficiency during climate variations	The application of indigenous knowledge in development programmes through the participation of women to reduce food insecurity can yield positive results
[30]	Quantitative	Malawi	A study commissioned to evaluate the effects of droughts and floods on rural households. The influence of safety nets on both protection and promotion effects	Policy options and reduction in food security. Livelihood promotion and -protection. Policy can compensate for market failures	Policy can compensate for market failuresEvery income transfer was invested in income-generating activities
[37]	Quantitative	*n* = 1800 (farm households) Ethiopia and South Africa	Both countries’ adaptation strategies and factors influencing the decision to adapt were analysed in this study	Common adaptation strategies applied by Ethiopia and South Africa included crops, tree plantations, the conservation of soil, changes to planting dates and irrigation methods	Access to information, markets and credit can be enabled by policymakers to support adaptation

Source: Authors’ own creation.

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
