# Peer review of "Drought Influences on Food Insecurity in Africa: A Systematic Literature Review"

_ijerph, 2020, doi:10.3390/ijerph17165897_

Round 1
Reviewer 1 Report
Drought influences on food insecurity in Africa: A synthesized literature review
This manuscript reviews the previous studies on climate change, food security, and drought-related issues in African countries. The idea itself is interesting, and authors seem to put forth the significant effort collecting, analyzing, and synthesizing the results. However, several issues must be addressed in order for this manuscript to be considered for publication.
Major comments:
- The primary concern I have with this review article is writing and English structure. This manuscript is very poorly written (I’m sorry I must say this). Please try to maintain follow, verbose, several adverbs, passive voice while writing a manuscript. To be very honest, I got lost several times while reading this piece of work. I would also kindly ask the authors either to follow British or American English properly.
- An abstract is so long. I would write in an active voice and simple sentence. Try to make abstract short, and concise.
- Minor comments:
- I would change the title of the manuscript, change “synthesised” to synthesized.
- Page 1-line 9: “Unfavourable” to “unfavorable.”
- Page 1-line 10: Put a comma (,) after climate variation,
- Page 1- line 19: define NVIVO
- So on.
Author Response
Reviewer 1: |
|
The primary concern I have with this review article is writing and English structure. This manuscript is very poorly written (I’m sorry I must say this). Please try to maintain follow, verbose, several adverbs, passive voice while writing a manuscript. To be very honest, I got lost several times while reading this piece of work. I would also kindly ask the authors either to follow British or American English properly. |
Done???? Services of the language editor has been sourced |
An abstract is so long. I would write in an active voice and simple sentence. Try to make abstract short, and concise. |
It has been reduced to 238 words |
I would change the title of the manuscript, change “synthesised” to synthesized. |
Done |
Page 1-line 9: “Unfavourable” to “unfavorable.” |
Done |
Page 1-line 10: Put a comma (,) after climate variation |
Done |
Page 1- line 19: define NVIVO |
Done |
Reviewer 2 Report
This is a well written article systematically surveying the last couple of decades of research on food insecurity in Sub Saharan Africa.
Given the imminent problems that the covid-19 pandemic, and the attempts to slow its spread are about to have on nutrition levels in many parts of Africa it is a timely baseline study against which new developments in coming months can be bench-marked.
Author Response
Thank you.
Reviewer 3 Report
- It is suggested to modify the last part of the Abstract (Lines 28-33; 35-37), which seems to give a laudatory opinion on the paper, overestimating the merits of the paper.
- At the lines 110-113, it is opportune to identify the difference between climate change adaptation and climate change mitigation, as the former indicates the set of actions to minimize the risk and the latter (mitigation) to contrast the causes of climate change (see Peres D. Adaptation strategies to climate change for water resources, In “Water Resources of Italy”(eds. G.Rossi and M Benedini), Spinger, 2020, 335-353).
- Revise the Table 1, as the syntheses of the articles are not homogeneous. It is opportune complete the table with additional information on the general topic: e.g. impacts of a specific drought, vulnerability to climate change, etc.)
- It is necessary to explain the difference among the papers included in the table 1 and the other papers, not listed in the table, but reviewed in the 4 sections which deal with the main themes of the research.
- It is required to revise the order of the sections 5,6,7 and 8, as it is not clear the reasons for splitting adaptation strategies (5) and resilience strategies (8). Also a logical sequence could be to examine first the effects of drought and vulnerability to food insecurity and after the strategies. Anyway it is suggested to change the sections 5,6,7, and 8 into subsections of section 4, i.e 4.2, 4.3, 4.4, 4.5.
- The following citations in the text are lacking of references: line 48 FAO(2017); line 102 United Nations, 1975; line 274 World bank 2003; line 347 Hygo, 2005-2015; Sendai 2005-2013.
Minor comments:
-line 42: change “line”
-45: check” prolonged and recurrence”
-68: modify “ prevalent”
-Table 1, art [28] last column, line 5 : correct specialist
-258: check “has been observed”
-268: check “plethora of authors”
-363: cancel 2018
-579: cancel 67 del Ninno
References :
Many titles of Journals should be modified in italic type: 7,11,,14,20,24,27, 29, 30, 37, 40,41, 50, 56, 57, 58, 63,64.
Some references should be completed: 35, 62,68
Author Response
It is suggested to modify the last part of the Abstract (Lines 28-33; 35-37), which seems to give a laudatory opinion on the paper, overestimating the merits of the paper. |
Done |
At the lines 110-113, it is opportune to identify the difference between climate change adaptation and climate change mitigation, as the former indicates the set of actions to minimize the risk and the latter (mitigation) to contrast the causes of climate change (see Peres D. Adaptation strategies to climate change for water resources, In “Water Resources of Italy” (eds. G.Rossi and M Benedini), Spinger, 2020, 335-353).
|
I have read the book chapter by Peres 2020, however, it is irrelevant in this literature review paper |
Revise the Table 1, as the syntheses of the articles are not homogeneous. It is opportune complete the table with additional information on the general topic: e.g. impacts of a specific drought, vulnerability to climate change, etc.)
|
Done |
It is necessary to explain the difference among the papers included in the table 1 and the other papers, not listed in the table, but reviewed in the 4 sections which deal with the main themes of the research.
|
Addressed under section 4.1 under “Study characteristics” |
It is required to revise the order of the sections 5,6,7 and 8, as it is not clear the reasons for splitting adaptation strategies (5) and resilience strategies (8). Also a logical sequence could be to examine first the effects of drought and vulnerability to food insecurity and after the strategies. |
Corrected |
Anyway it is suggested to change the sections 5,6,7, and 8 into subsections of section 4, i.e 4.2, 4.3, 4.4, 4.5. |
Corrected |
The following citations in the text are lacking of references: 347 Hygo, 2005-2015; Sendai 2005-2013. |
It has been added |
-line 42: change “line” |
Corrected |
-45: check” prolonged and recurrence” |
Corrected |
-68: modify “ prevalent” |
Corrected |
-Table 1, art [28] last column, line 5 : correct specialist |
Corrected |
-258: check “has been observed” |
Corrected |
-268: check “plethora of authors” |
Corrected |
-363: cancel 2018 |
Corrected |
-579: cancel 67 del Ninno |
Corrected |
References : |
|
Many titles of Journals should be modified in italic type: 7,11,,14,20,24,27, 29, 30, 37, 40,41, 50, 56, 57, 58, 63,64. |
Corrected |
Some references should be completed: 35, 62,68 |
|
Reviewer 4 Report
The topic seems to me truly relevant and of great interest at the present time, however I see fundamentally methodological and organizational deficiencies that must be improved.
- The wording of the abstract should be revised: it is confusing and repetitive
- The manuscript structure does not need to be explained in the introduction (lines 127-130) since it must necessarily comply with the structure of a scientific review article.
- The results are presented "raw", the authors explain that they have used a computer program to encode the information of the selected articles. However, they do not show any maps, diagrams, charts, tables, etc. obtained with the Nvivo program that encodes, classifies and relates the information extracted from documentary sources.
- The numbering of the epigraphs from "results" onwards must be modified, since if the following sections are linked to the four categories that the authors select to explain the results, they must be sub-epigraphs of results, that is, 4.1; 4.2; 4.3 .... Also, I would add "discussion" in section 4.
- The conclusions should be related to the objectives more clearly.
Author Response
The wording of the abstract should be revised: it is confusing and repetitive |
Revised |
The manuscript structure does not need to be explained in the introduction (lines 127-130) since it must necessarily comply with the structure of a scientific review article. |
Removed |
The results are presented "raw", the authors explain that they have used a computer program to encode the information of the selected articles. However, they do not show any maps, diagrams, charts, tables, etc. obtained with the Nvivo program that encodes, classifies and relates the information extracted from documentary sources. |
Included in the text |
The numbering of the epigraphs from "results" onwards must be modified, since if the following sections are linked to the four categories that the authors select to explain the results, they must be sub-epigraphs of results, that is, 4.1; 4.2; 4.3 .... Also, I would add "discussion" in section 4. |
Already modified |
The conclusions should be related to the objectives more clearly. |
Sorted |
Round 2
Reviewer 1 Report
I am pleased with the progress this manuscript has made. Still, it warrants spelling check and English language, for example in a table we can clearly see 2/3 different font types and sizes. Please pay attention on how to clearly present your results. Thanks